



# 1 Increasing Frequency in Off-Season Tropical Cyclones and its
# 2 relation to Climate Variability and Change

**José J. Hernández Ayala[1] and Rafael Méndez-Tejeda[2]**
[1] Department of Geography, Environment & Planning, Climate Research Center, Sonoma State
University, California, USA. jose.hernandezayala@sonoma.edu
[2] Research Laboratory in Atmospheric Science, University of Puerto Rico at Carolina, Puerto Rico.
P. O. Box 4800, 00984, Carolina, Puerto Rico. rafael.mendez@upr.edu
**Correspondence**: José J. Hernández Ayala (jose.hernandezayala@sonoma.edu)
**Abstract.** This article analyzes the relationship between off-season tropical cyclone (TC)
frequency and climate variability and change for the Pacific and Atlantic Ocean basins. TC track
data was used to extract the off-season storms for the 1900-2019 period. TC counts were
aggregated by decade and the number of storms for the first six decades (pre-satellite era) was
adjusted. Mann-Kendall non-parametric tests were used to identify trends in decadal TC counts
and multiple linear regression models (MRL) were used to test if climatic variability or climate
change factors explained the trends in off-season storms. MRL stepwise procedures were
implemented to identify the climate variability and change factors that explained most of the
variability in off-season TC frequency. A total of 713 TCs were identified as occurring earlier or
later than their peak seasons, most during the month of May and in the West Pacific and South
Pacific basins. The East Pacific (EP), North Atlantic (NA) and West Pacific (WP) basins exhibit
significant increasing trends in decadal off-season TC frequency. MRL results show that trends
in sea surface temperature, global mean surface temperature, and cloud cover explain most of
the increasing trend in decadal off-season TC counts in the EP, NA, and WP basins. Stepwise
MLR results also identified climate change variables as the dominant forces behind increasing
trends in off-season TC decadal counts, yet they also showed that climate variability factors like
El Niño-Southern Oscillation, the Atlantic Multidecadal Oscillation, and the Interdecadal Pacific
Oscillation also account for a portion of the variability.
**Keywords:** Tropical Cyclones; Hurricane Season; Climate Variability; Climate Change

## 1. Introduction
Increasingly, scientific evidence has shown a link between tropical cyclones (TC) and global
warming, especially following the dramatic rise in both the intensity and frequency of storms during
the first two decades of the present century (Goldenberg et al., 2001; Holland and Webster, 2007).
Scientific studies (Landsea, 2005; Emanuel, 2005; Trenberth and Shea, 2006; Trenberth, 2007) are
not in agreement as to whether sea surface temperatures have a measurable effect on the frequency
of tropical cyclones and other studies (Camargo and Sobel, 2005; Nogueira and Kim, 2007; Mahala
et al., 2015; Zhao et al., 2018] have evaluated cyclonic activity on a time scale longer than
interannual and have associated it with variability in the El Niño Southern Oscillation (ENSO), the
Atlantic Multidecadal Oscillation (AMO) and the Interdecadal Pacific Oscillation (IPO). However,
little is known about the changes in the frequency of off-season TCs, storms that occur before and
after the peak TC season months, and their connections to climate variability and change.
A number of recent papers (Wang and Lee, 2008; Knutson et al., 2010; Emanuel, 2013) have
documented global increases in the proportion of very intense cyclones as well as latitudinal trends
in maximum tropical cyclone (TC) intensity, which are consistent with future climate projections. A
detailed review of the behavior of TCs (Walsh et al., 2016) concluded that it remains uncertain
whether past changes in TC activity have exceeded the variability expected from natural causes,



while concerns remain about the temporal homogeneity of the best record (Landsea et al., 2006;
Mann et al., 2007). Another study (Mann et al., 2009) found that recent increases in the frequency
of intense TCs in the North Atlantic (NA) were the product of reinforcing effects, such as La Niña-
like climate conditions and relative tropical Atlantic warming. Yet, no study has focused on
examining changing trends in off-season TC frequency and its relation to natural variability or
climate change.
A synthesis (Christensen et al., 2013) of the then-available regional projections of future TC
climatology for 2081–2100 in relation to 2000–2019, for a business as usual emissions scenario
similar to A1B, shows that worldwide the consensus projection was for decreases in TC numbers by
approximately 5–30%, increased frequency of Category 4 and 5 storms between 0 and 25%, an
increase by a small percentage in the typical maximum intensity of life, and an increase in TC
rainfall amounts by 5–20%. Nevertheless, it is clear that there is great uncertainty about these
projections. Such projections do not consider changes in off-season TC development in any of the
basins where TCs form
Several climatic reconstructions have been performed (Bradley et al., 2006; Mann et al., 2009)
using proxy data by collecting sediments from the impact of hurricanes in the period 500–1850 and
then calculated estimates from the statistical model of the activity of tropical cyclones based on
modern instrumental weather indexes for the period (1851–2006). In analyzing these results and
comparing them with the cyclone seasons fixed by the World Meteorological Organization, the
hurricane season (tropical depressions, tropical storms and hurricanes) in the Atlantic Ocean was
fixed as June 1 to November 30 in 1960, yet we observe a significant variability in off-season TC
occurrence before/after the hurricane season after the 1960s.
The formation of the extratropical storm Andrea on May 20, 2019 marks the decade of 2010 as that
with the greatest number of tropical cyclones in the Atlantic Ocean before or after the hurricane
season dates established by official bodies like the World Meteorological Organization (WMO) and
the National Oceanic and Atmospheric Administration (NOAA). The frequency of TCs in the North
Atlantic basin has been found to be influenced by fluctuations in teleconnections such as ENSO and
AMO (Trenberth et al., 2006; Nogueira et al., 2013). However, human-induced climate change
manifested as higher sea surface temperatures (SST) and increasing evaporation rates in the tropical
and sub-tropical North Atlantic basin could also be related to the higher frequency of off-season
tropical or extratropical cyclone occurrences in more recent decades. That increasing trend in SSTs
in the Atlantic and other ocean basins and its relation to out off-season TC occurrences during the
last century has not been thoroughly examined by the scientific community.
This study aims to determine if off-season TCs have increased in their frequency since the 1900 and
if that increment in the number pre and post off-season storms could be associated with normal
climatic variability or climate change. The total number of out off-season TCs per decade for the
North Atlantic (NA), West Pacific (WP), East Pacific (EP) and South Pacific (SP) ocean basins
where analyzed to determine if any of the basins experienced an increase in the number of off-
season tropical/extratropical cyclones over time that could be associated to climatic variability or
climate change. The Indian Ocean basins were not included in this analysis due to limited data
availability. Ocean basins that were found to have statistically significant trends were then analyzed
further with multiple liner regression models (MRL) and regression stepwise procedures to
determine if climate variability or change factors could explain increasing trends in off-season TC
frequency over time.





## 2. Data

Six-hourly TC track data for all storms across all ocean basins were obtained from the International
Best Track Archive for Climate Stewardship (IBTrACS) (Knapp et al., 2018) and all TCs that
occurred at or after 1900 were extracted. The TC tracks were then then extracted for the northern
hemisphere basins that include the East Pacific (EP), the North Atlantic (NA) and the West Pacific
(WP) and for the southern hemisphere basin in the South Pacific (SP) (Fig 1a). The off-season TCs
were then aggregated by decades in order to identify decadal variability or trends in total storm
counts at the individual basin scales. Off-season TCs were defined as storms that occurred in the
three months before and after the six-month period of peak cyclone activity in the basin.
The monthly frequency of TCs for each basin were analyzed for the entire period and based on that
analysis we determined that off-season TCs that occurred during the three months (Mar-Apr-May)
before June 1st were pre-season and the three months (Dec-Jan-Feb) after November 30th were post-
season in the northern hemisphere basins (NA, EP and WP) In the southern hemisphere, the three
months before (Aug-Sep-Oct) November 1st were classified as pre-season and the three months
(May-Jun-Jul) after April 30th were classified as post off-season in the southern hemisphere basins
(SP) (Fig 1b). Pre-season and post-season decadal time-series for the Northern/Southern hemisphere
and individual basins were then constructed to calculate the total number of TCs per-decade from
1900 to 2019.
The climate variability indexes of ENSO (Niño 3.4), AMO (Trenberth et al., 2019) and IOD were
respectively obtained from the National Oceanic and Atmospheric Administration (NOAA)
Physical Sciences Lab, the National Center for Atmospheric Research, and the Australian
Government Bureau of Meteorology. The IPO index was obtained from the NOAA Physical
Sciences Laboratory (Henley et al., 2015). The variables associated with anthropogenic climate
change used in this study were sea surface temperature (SST), global mean surface temperature
(GMST) and cloud cover (CC). SST data were obtained from the HadISST1 1° reconstruction,
GMST data were accessed from the GISTEMP v4 and CC data was acquired from the ICOADS
v2.5, all for the 1900-2019 period. A decadal average was calculated for all of the climate
variability and change variables in order to use them as predictors of decadal TC total counts (Table
126 1).

**Table 1. Tropical cyclone, climate change and variability variables used in this study.**

|  | Abbreviations | Units |
| --- | --- | --- |
| Tropical Cyclone Counts | TCs | Decadal Total Counts |
| **Climate Change Variables** |  |  |
| Sea Surface Temperature | SST | ° |
| Global Mean Surface Temperature | GMST | ° |
| Cloud Cover | CC | Oktas |
| **Climate Variability Variables** |  |  |
| El Niño Southern Oscillation | ENSO 3.4 | °SST anomalies index |
| Interdecadal Pacific Oscillation | IPO | °SST anomalies index |
| Atlantic Multi-decadal Oscillation | AMO | °SST anomalies index |








## 3. TC Adjustment Method

TC counts before 1966 (pre-satellite era) are incomplete (Mann et al., 2007; Landsea, 2007) since a lot of storms that didn't make landfall weren't recorded, so in order to make any comparisons between the earlier and later decades, the series for each basin need to be adjusted accordingly. The average landfall percentage of TCs were calculated for the periods 1900-1965 (pre-satellite) and 1966-2019 (satellite era and new TC monitoring technologies available) in order to determine the share of storms that made landfall in both periods. The percentage of landfalling TCs is expected to be higher in the 1900-65 period since a higher number of storms that remained over the ocean were not reported, so the landfall percentage of the pre-satellite period is then adjusted so that it matches the 1966-2019 post-satellite period.

To obtain the estimated number of missing TCs for the 1900-65 period, the number of total storms in the pre-satellite period is increased until its landfall percentage is equal to the one in the post satellite era. The total number of additional TCs that resulted in the landfall percentages between the two periods to be the same or near equal are then divided by the 7 decades of the pre-satellite era and then the number of extra storms for each decade is multiplied by the percentage of off-season storms for each basin and that resulting number is then added then to each of the individual decades between 1900 and 1969. In a previous study (Landsea, 2007), this method was applied to adjust TC counts in the North Atlantic to determine if the basin has experienced an increasing trend in annual TC frequency since the 1900, and its results show that after adjusting the tropical storm counts no trends were found.

Here we show how this TC series adjustment method was applied to the total TC count for the NA basin for the 1900-2019 period . First, we calculate the landfall percentage for the pre-satellite period 1900-65 by dividing the number of landfalling TCs (LFTCs) with the total number of storms (TTCs) and multiply by 100 to get the landfall percentage, check the equations below:

$$
\begin{aligned}
&(LFTCs \,/TTCs) \,*\, 100 \\
&Example: (479/610) * 100 \,=\, 78.5\%
\end{aligned}
\tag{1}
$$

Then calculate landfall % for the period post-satellite period 1966-2019,

$$
\begin{aligned}
&(LFTCs \,/TTCs) \,*\, 100 \\
&Example: (583/844) * 100 \,=\, 69.1\%
\end{aligned}
\tag{2}
$$

Then artificially increase the number of TCs (+83 for the NA basin) until the landfall % of the 1900-65 period is equal to landfall % of the 1966-2019:

$$
\begin{aligned}
&LFTCs \,/\, (TTCs \,+\, AddTCs) \,*\, 100 \\
&Example: 479/ (610 + 83) * 100 \,=\, 69.1\%
\end{aligned}
\tag{3}
$$

Then calculate the percentage (OffP) of off-season TCs (OffTCs) by dividing it by total number of TCs:

$$
\begin{aligned}
&(OffTCs \,/TTCs) \,*\, 100 \\
&Example : (67/1454) * 100 \,=\, 4.61\%
\end{aligned}
\tag{4}
$$

Then divide additional TCs (83) by the number of decades between 1900 and 1969 (7) and then multiply by the off-season TC percentage (.0461)






$$(AddTCs/Decades) * OffP$$
$$Example: (83/7) * .0461 = 0.54 \tag{5}$$

In the case of the NA, we determined that by using the above TC series adjustment method the basin
would get an additional 0.54 off-season TCs for each of the seven decades that go from the 1900 to
1969. Finally, the additional 0.54 TCs per decade will be divided between pre and post off-season
TCs by multiplying the added storms with the respective percentage of pre/post off season cyclones:

$$DecOffTCs * Percentage/Post Season$$
$$Example: 0.54/0.62 = 0.33 \ and \ 0.54/0.38 = 0.21 \tag{6}$$

The pre off-season decades of the NA basin before 1970 will get an additional 0.33 TCs and the
post off-season decades will get 0.21 more storms. This off-season TC adjustment method was
applied to the other five basins.
**4. Statistical Methods & Models**
Mann-Kendall (MK) tests for trends (Mann, 1945; McLeod, 2005) were applied to all the off-
season TC decadal series for all basins in order to determine if the frequency of storms has increased
or decreased over time. This test has the advantage of not assuming any special form for the
distribution function of the data, while having a power nearly as high as their parametric equivalents
and that is why its use is highly recommended by the World Meteorological Organization (Hipel and
McLeod, 2005). The Mann-Kendall rank statistic t is calculated according to:

$$S = \sum_{k=1}^{n-1} \sum_{j=k+1}^{n} \text{sgn}\,(X_j - X_k) \tag{1}$$

with

$$\text{gn}\,(x) = \begin{cases} 1 & \text{if} x > 0 \\ 0 & \text{if} x = 0 \\ -1 & \text{if} x < 0 \end{cases} \tag{2}$$

The mean of $S$ is $E[S] = 0$ and the variance $\sigma^2$ is

$$\sigma^2 = \{n(n-1)(2n+5) - \sum_{j=1}^{p} t_j(t_j - 1)(2t_j + 5)\}/18 \tag{3}$$

where p is the number of the tied groups in the data set and $t_j$ is the number of data points in the $_{j}th$
tied group. The statistic $S$ is approximately normal distributed provided that the following Z-
transformation is employed:

$$Z = \begin{cases} \dfrac{S-1}{\sigma} & \text{if} \quad S > 0 \\ 0 & \text{if} \quad S = 0 \\ \dfrac{S+1}{\sigma} & \text{if} \quad S > 0 \end{cases} \tag{4}$$

The statistic S is closely related to Kendall's $\tau$ as given by:





$$\tau = \frac{S}{D} \tag{5}$$

where

$$D = [\frac{1}{2}n(n-1) - \frac{1}{2}\sum_{j=1}^{p} t_j(t_j-1)]^{1/2}[\frac{1}{2}n(n-1)]^{1/2} \tag{6}$$

The decadal series that were then found to have a significant trend based on the MK results were
then furtherly analyzed by applying a series of multiple linear regression models (MLR). MLR were
used to model the decadal count of off-season TCs for basins that showed increasing or decreasing
trends in storm numbers to test if covariates associated with climatic variability and climate change
explained off-season TC frequency. MLR attempts to model the relationship between two or more
explanatory variables and a response variable by fitting a linear equation to observed data. The
notation for the model deviations is:

$$y_i = \beta_0 + \beta_1 x_{i1} + \beta_2 x_{i2} + \cdots \beta_p x_{ip} + \varepsilon_i \text{ for } i = 1,2,\dots n \tag{1}$$

Three different MLR models were run for each off-season TC series that exhibited a statistically
significant trend, one MLR model with the climate change variables (SST, GMST & CC) as
predictors, another model with just the climate variability factors (ENSO, AMO & IOD) and a final
model with all of the variables included. Then the three MLR models (pre-season, post-season and
off-season) were run for each of the basins with increasing trends in off-season TCs, the best models
(highest adjusted R-squared and lowest p-value) were then selected for each of the series. The MLR
models were run in The R Project for Statistical Computing using the biglm package.
Finally, stepwise selection MLR models were used to identify the climate variability or change factors
making the most statistically significant contributions to off-season increasing TC frequency. Here
we use stepwise selection which is a combination of the forward and backward procedures where you
start with no predictors, then sequentially add the most contributive predictors. After adding each new
variable, it removes the variables that no longer provide an improvement in the model fit (James et
al., 2014; Bruce and Bruce, 2017). The MLR and stepwise for the off-season TC count series for each
of the basins with significant increasing trends were run in The R Project for Statistical Computing
using the MASS package (Venables and Ripley, 2002).

## 5. Results & Discussion

When analyzing the number of TCs for all basins for the 1900-2019 period we found that 713
off-season storms occurred during that time, most during the months of May (NH pre-season and SH
post-season) with 430 and December (NH post-season) with 341 (Figure 1a, 1b). When looking at
the count of off-season TCs per basin we found that as expected the West Pacific (611) and South
Pacific (85) accounted for 81.3% of all off-season storm occurrences. When grouping the basins
between northern and southern hemispheres, we find that 89% of all off-season TCs occurred north
of the equator for the 1900-2019 period (Figure 1a, 1b). The North Atlantic and East Pacific basins
were found to be the ones with the lowest numbers of off-season TCs when compared to the other
two Pacific basins.

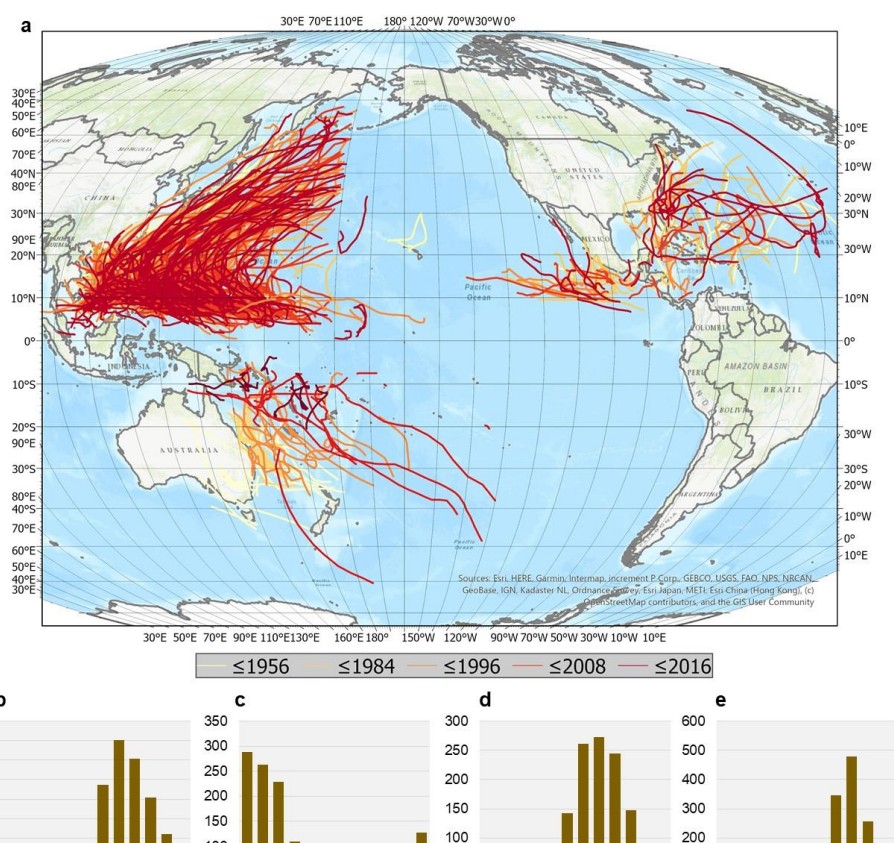

**Figure 1.** Tracks of all out off-season TCs (a) and the number of storms per month for the WP (b),
SP (c), EP (d) and NA (e) basins for the 1900-2019 period.

As shown in Figure 2, even after adding the estimated missing storms before the 1970 decade,
most basins experienced their highest number of out off-season TCs (pre or post) in decades at or
after 1960-69. The 1960-69 decade for the northern hemisphere basins (WP, NA and EP) was found
to be the one with the highest number of pre off-season TCs with 69 and the 1950-1959 decade was
identified as the one with the most post off-season storms with 68 (Figure 2a, 2b, 2c and 2d). When
examining TC counts for all basins individually, we found that the NA and EP basins had their most
active decades after 1970 and that the WP and SP basins experienced their highest storm count decade
after 1960 (Figure 2c, 2d). It is important to note that these results already reflect the additional TCs
that were added to the pre-satellite era.

258



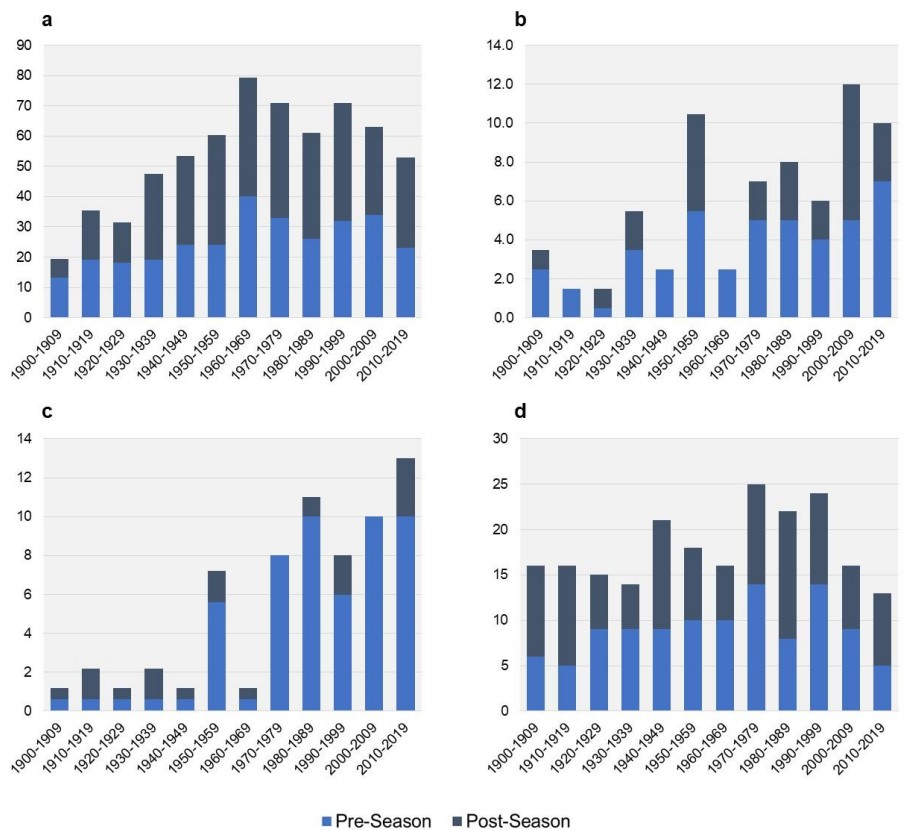

Pre-Season ■ Post-Season

**Figure 2.** Adjusted decadal count of all observed and estimated off-season TCs for the WP (a), NA (b), EP (c) and SP (d) ocean basins for the 1900-2019 period.

The Mann-Kendall non-parametric tests for trends for all basins show that three basins exhibited statistically significant increasing trends in adjusted decadal off-season TC counts for the 1900-2019 period (Table 2). The basins with statistically significant increasing trends were the EP (pre and off-season), NA (pre, post and off-season) and the WP (pre, post and off-season). The EP basin shows an increasing trend in pre- and off-season TCs that is more evident from the 1950s to the present (Figure 2d), while the increasing trend in the NA basin can be observed from the 1970s to 2019.

The increasing trend in off-season TCs in the WP basin is more evident from the 1900 to 1969 (Figure 2a), which was during the pre-satellite era where missing TCs were added to the series. However, no trend is found in the WP basin if the decadal counts are analyzed from the 1970s to the present. In the post-satellite era in the WP basin, the 1990-1999 decade was identified as the one with most off-season TCs, however the two following decades exhibited a decreasing trend. The EP and NA basins show significant increasing trends in off-season TC counts (Table 2). Opposites to the trends identified in the WP basin, the EP and NA also show increasing decadal counts after the 1960s and 1970s decades. The SP basin also exhibited a positive Tau coefficient, yet it was statistically insignificant (Figure 2d).






**Table 2. Results of Mann-Kendall trend tests for the 1900-2019 period for all ocean basins where TCs form.**

| Pre TCs | Tau S | P-value | Post TCs | Tau S | P-value | Off TCs | Tau  S | P-value |
|---------|-------|---------|----------|-------|---------|---------|--------|---------|
| EP | **0.746** | **0.002** | EP | 0.098 | 0.723 | EP | **0.679** | **0.004** |
| NA | **0.572** | **0.015** | NA | **0.485** | **0.042** | NA | **0.554** | **0.016** |
| SP | 0.048 | 0.889 | SP | 0.015 | 1.000 | SP | 0.061 | 0.836 |
| WP | **0.554** | **0.016** | WP | **0.485** | **0.034** | WP | **0.534** | **0.019** |

**Significant trends in bold.**


MLR models were run on the basins that exhibited statistically significant (< 0.05) increasing
trends in decadal total off-season TC counts over time and here we report the best models for each of
the series. The MLR results show that the statistically significant increasing trends in TC frequency
for the EP (pre and off-season) and WP basins is best explained by climate change factors SST, GMST
and CC at the 0.05 significance level (Table 3). Climate change factors accounted for 56% (pre-
season) and 52% (off-season) of the increasing trend in TC counts for the EP basin. In the WP basin
climate change factors explained 55% (pre-season), 64% (post-season) and 68% (off-season) of the
trends in off-season TCs**.** Increasing trends in SSTs, GMST and moisture (CC) outside of the prime
months of tropical storm development could promote more optimum conditions for higher off-season
TC occurrences (Klozback, 2006; Hansen et al., 2010).

**Table 3. Best multiple linear regression models (MLR) for basins with statistically significant increasing trends in off-season TCs.**

| Model | Multiple R-squared | Adjusted R² | Factors | p-value |
|-------|--------------------|-------------|---------|---------|
| EP pre-season | 0.682 | 0.563 | SST, GMST & CC | 0.021 |
| EP off-season | 0.653 | 0.522 | SST, GMST & CC | 0.030 |
| NA pre-season | 0.622 | 0.481 | SST, GMST & CC | 0.041 |
| NA post-season | 0.687 | 0.427 | ENSO & AMO | 0.130 |
| NA off-season | 0.552 | 0.384 | SST, GMST & CC | 0.070 |
| WP pre-season | 0.673 | 0.551 | SST, GMST & CC | 0.020 |
| WP post-season | 0.742 | 0.645 | SST, GMST & CC | 0.000 |
| WP off-season | 0.774 | 0.689 | SST, GMST & CC | 0.005 |


The climate variability factors (ENSO, AMO & IOD) did not exhibit statistically significant
relationships with increasing off-season TC counts, which shows that natural variability does not
explain the incrementing number of storms in the EP and WP basins.  MLR model results for the
NA basin also showed the climate change variables accounting for 48% (pre-season) and 38% (off-
season) of the increasing trend in TCs (Table 3). However, the MLR model results for the post-
season months in the NA basin showed that the climate variability variables (ENSO & AMO)
accounted for 42% of the increasing trend in TCs, yet the model was not found to be statistically
significant. It is well known that cold phases of ENSO (La Niña) and warm phases of AMO tend to
be associated with higher TC frequency in the North Atlantic ocean (Tang and Neelin, 2004;
Briggs, 2008) and this could explain why those teleconnections were  found to have the most
significant influence on post-season TC frequency in the NA basin. Yet, it is important to note that
in most basin series, including the NA, climate change variables explained more of the off-
season TC increasing trend than the climate variability factors.





Stepwise MLR model results showed that climate change factors (SST, GMST & CC) were among
the selected variables that explained most of the increasing trend in off-season TCs for all basins
analyzed (Table 4). In the EP basin, SST, ENSO, and CC, accounted for 69% (pre-season) and 65%
(off-season) of the increasing trend in TCs. In the NA basin, the stepwise procedure selected CC as
the sole climate change factor that explained 52% (pre-season) and 40% (off-season) of the rising
frequency in TC counts. However, CC & AMO were selected as the variables that explained (43%)
most of the variability in TC frequency during the post-season months in the NA basin. Stepwise
procedure results for the WP basin show that climate change and variability factors were selected as
the best predictors of TC frequency, with GMST and CC accounting for 57% (pre-season), CC,
GMST, ENSO and IPO explaining 72% (post-season) and 74% (off-season) of the variability of
TCs.

**Table 4. Stepwise multiple linear regression models (MLR) for basins with statistically significant increasing trends in off-season TCs.**

| Model | R-squared | Adjusted R² | Factors | p-value |
|---|---|---|---|---|
| EP pre-season | 0.777 | 0.694 | SST, ENSO & CC | 0.005 |
| EP off-season | 0.747 | 0.652 | SST, ENSO & CC | 0.008 |
| NA pre-season | 0.569 | 0.526 | CC | 0.004 |
| NA post-season | 0.687 | 0.427 | CC & AMO | 0.098 |
| NA off-season | 0.460 | 0.406 | CC | 0.015 |
| WP pre-season | 0.655 | 0.578 | GMST & CC | 0.008 |
| WP post-season | 0.826 | 0.726 | CC, GMST, ENSO & IPO | 0.008 |
| WP off-season | 0.839 | 0.747 | CC, GMST, ENSO & IPO | 0.006 |


319       The EP experienced a steady increase in off-season TC total counts from 1900 to 2019 at a rate
of 1.1 additional storms per decade. The decadal off-season total TC count series for the EP basin
closely resembles the increasing trend in average SSTs and CCs (Fig 3a, 3c). The correlation between
off-season TCs in the EP basin and ENSO is not as clear as the one between SST and CC, with some
mostly warm ENSO decades like the 1990-1999 exhibiting lower storm counts and other periods with
cooler phases dominating showing a higher number of cyclones. When SST patterns for areas in the
EP basin where TCs develop are examined over time, we find that most tropical/sub-tropical ocean
waters have experienced a statistically significant increasing trend in ocean surface temperatures from
1900 to 2019 (Fig 3d). Similar to other studies (Hansen et al., 2010), we find that the EP tropical
ocean surfaces have increased by 0.051 degrees C° per decade. When CC patterns are examined, we
find that it has also experienced a statistically significant increasing trend in some areas in the EP
basin (Fig 3e).

332       The decadal off-season total TC count series for the NA basin closely resembles the increasing
trend in average SSTs and CCs (Fig 4a, 4c). The NA decadal series shows a steady increase in off-
season TC total counts from 1900 to 2019 at a rate of 0.7 additional storms per decade and an SST
increasing trend of 0.055 C° per decade. Both the average decadal SST and CC series coincide with
the peaks and valleys in off-season TC counts for the NA basin, with the 1950-1959 showing a high
number of storms associated with high mean SSTs and CCs while the drop in storm counts in the
1960-1969 decade matches a drastic drop in ocean surface temperatures (Figures 4a, 4c). Even though
average SSTs increase to 0.135 C° per decade from 1970 to 2019, off-season TC total counts went
down in the 1990-1999 and 2010-2019 decades, with the decade in between (2000-2009) exhibiting
the highest number of off-season TCs (14) of all decades examined. However, it is important to note
that 5 out of the 6 decades with the most off-season TCs in the NA basin occurred after the 1970s.

When North Atlantic SSTs are examined in areas where TCs form, we found that ocean surface
temperatures have increased at a rate of 0.055 degrees C° per decade for the off-season months of
Dec-March (Fig 4d). When CC patterns are  examined, we find that it has also experienced a
statistically significant increasing trend of 0.06 oktas (eighths of the sky that are covered in clouds)
per decade in the North Atlantic basin since the 1900 (Figure 4e). If the NA pre/post off-season series
is modified to begin in the 1960s, we find that SSTs have increased at a decadal rate of 0.082 C° per
decade  at a rate of 1.2 additional storms per decade. Overall, these results suggest that increasing
trends in SSTs, which also drive increasing trends in evaporation rates associated with high CCs, are
the physical mechanisms behind most of the recent increase in the total number of out of season TCs
in the NA basin. The correlation between off-season TCs in the NA basin and AMO is not as clear as
the one between SST and CC, with some warm AMO phases between 1930-1959 exhibiting lower
storm counts while some cooler phases (1970-89) showing a higher number of cyclones.

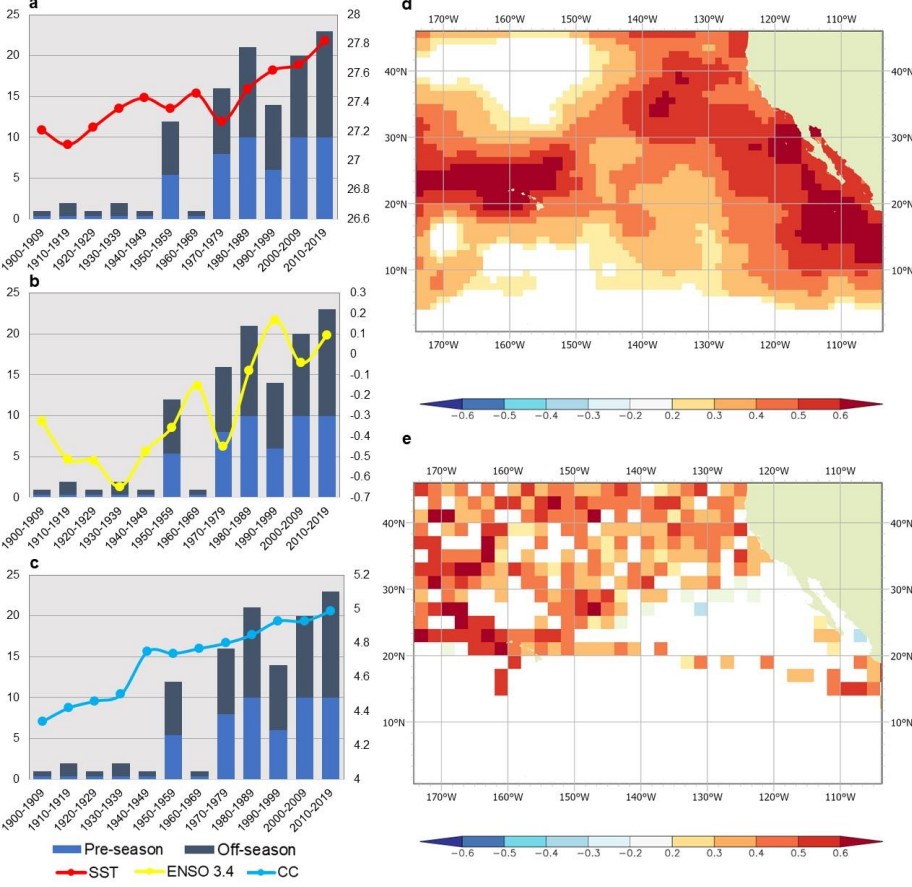


**Figure 3.** Decadal TC counts for the EP off-seasons and decadal average SSTs (a), decadal TC counts for
the EP off-seasons and decadal average ENSO 3.4 (b), decadal TC counts for the EP off-seasons and
decadal average Correlation between Time and Dec-May averaged CC (c), correlation between Time and
Dec-May averaged SST (C°) for the 1900-2019 period (d) and correlation between Time and Dec-May
averaged CC (oktas) for the 1900-2019 period (e).

363

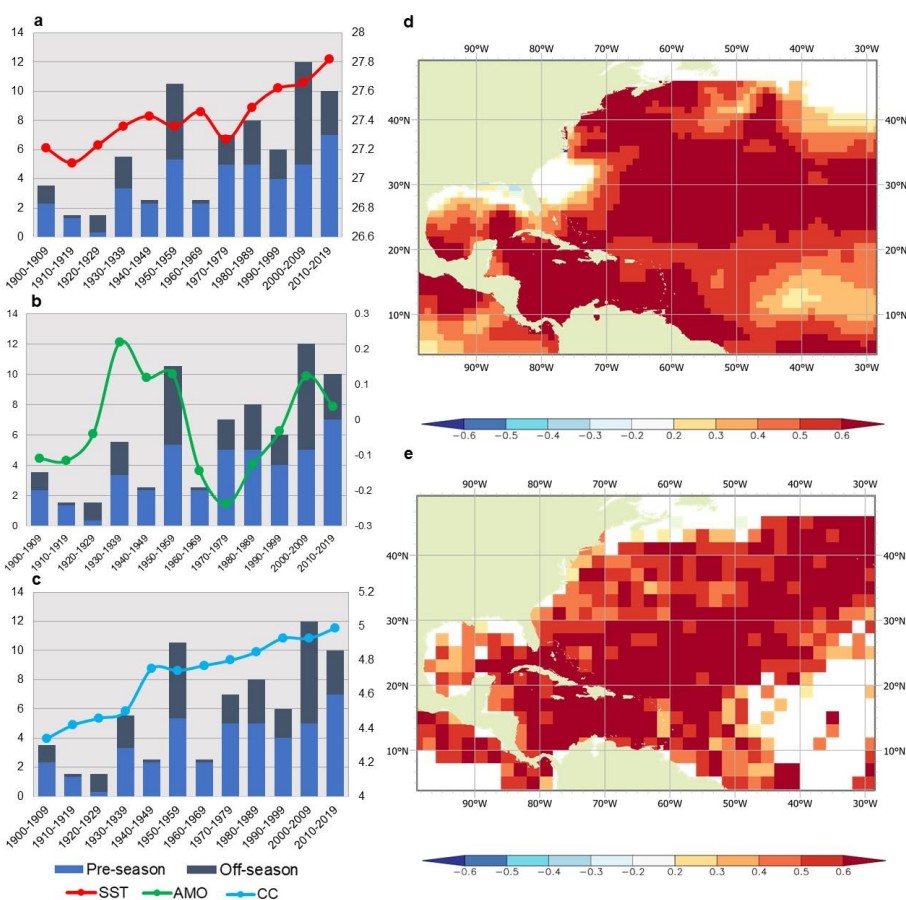

364

**Figure 4.** Decadal TC counts for the NA off-seasons and decadal average SSTs (a), decadal TC counts for the
NA off-seasons and decadal average AMO (b), decadal TC counts for the NA off-seasons and decadal average
Correlation between Time and Dec-May averaged CC (c), correlation between Time and Dec-May averaged
SST (C°) for the 1900-2019 period (d) and correlation between Time and Dec-May averaged CC (oktas) for the
1900-2019 period (e).

The decadal off-season total TC count series for the WP basin closely resembles the increasing
trend in GMST (Fig 5a). However, the WP basin experienced the highest count of off-season TCs in
the 1960-69 decade, not in the more recent decades like the EP and NA basins. More importantly, if
trend analysis for off-season TC counts is done from 1960-2019 in the WP basin, we find no
statistically significant increasing or decreasing trend. However, it is important to note that four out
of the five decades with most off-season TCs in the WP basin occurred after 1960. However, the
2010-2019 decade was identified as the period with the lowest total number of off-season TCs even
though increasing trends in mean SST, GMST and CC continued (Fig 5a, 5d and 5e). The decreasing
number of off-season TCs in the last two decades coincided with a negative phase of the IPO, which
suggests that TC frequency in the WP basin is influenced by fluctuations in the IPO (Fig 5c), whose
recent negative phase since 1998 resembles La Niña–like SST anomaly patterns (Zhao et.al, 2018).
Even though most of the variability in off-season TC frequency in the WP basin can be explained by
climate change trends in GMST, SST and CC, the rest of the variance in TCs is account by fluctuations
in the IPO and ENSO teleconnections.



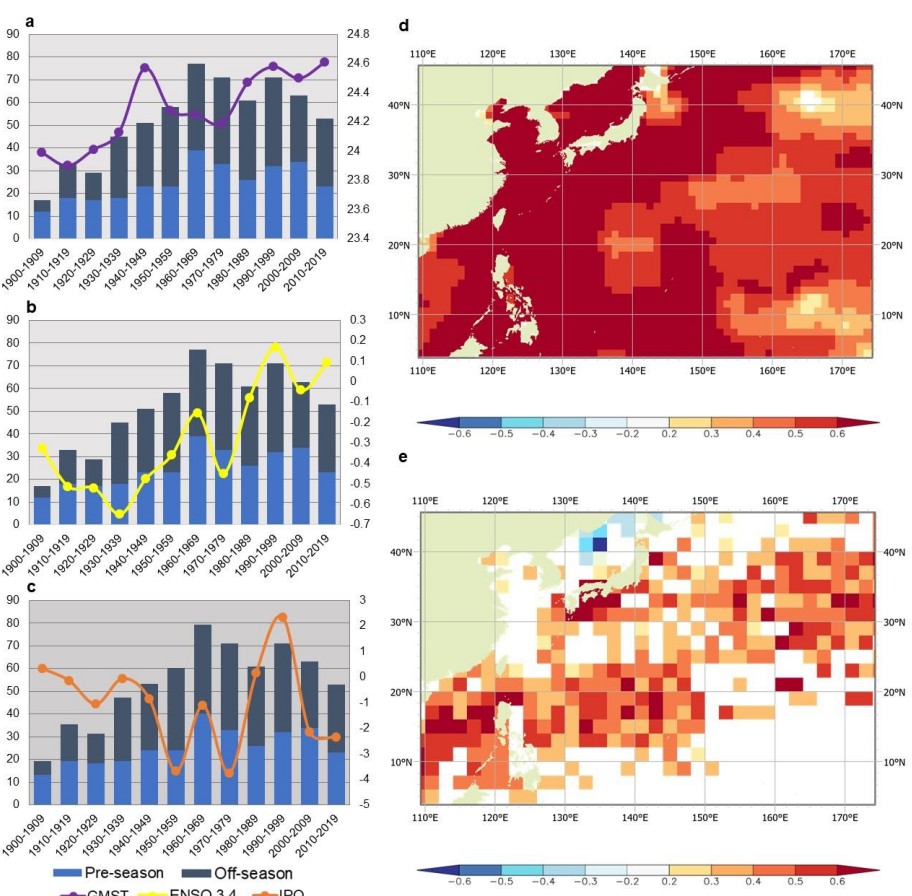

385
386
**Figure 5.** Decadal TC counts for the WP off-seasons and decadal average SSTs (a), decadal TC counts for the
WP off-seasons and decadal average AMO (b), decadal TC counts for the WP off-seasons and decadal average
Correlation between Time and Dec-May averaged CC (c), correlation between Time and Dec-May averaged
SST (C°) for the 1900-2019 period (d) and correlation between Time and Dec-May averaged CC (oktas) for the
1900-2019 period (e).

## 6. Summary and concluding remarks

The frequency of TCs that developed outside of their prime season months were analyzed to
determine if trends in higher storm totals in the Pacific and Atlantic Ocean basins were associated
with natural variability, climate change or both. Adjusted off-season decadal TC total counts for six
ocean basins were analyzed for the 1900-2019 period in order to determine if the number of storms
have been increasing over time. Mann-Kendall tests for trends were done and the basins that exhibited
statistically significant increasing trends were then furtherly analyzed using multiple linear regression
models and stepwise procedures to determine if those trends could be explained by fluctuations
associated with climate variability, climate change trends or a combination of both.

The main results of this study suggest that decadal total off-season (pre/post) TC counts have
significantly increase over time since the 1900 in the East Pacific (EP), North Atlantic (NA) and West
Pacific (WP) basins. The EP and NA basins exhibited statistically significant increasing trends even
if the analysis was done from the 1960s instead of the 1900. The WP basin showed an overall





increasing trend in the total number of off-season TCs per decade, yet if the analysis is done from the 1960s to the present, no statistically significant increasing trend is found. However, the three basins that reflected an overall increase in decadal off-season TC frequency had their most active decades after the 1970s.

Results from the best multiple linear regression (MLR) models show that the increasing decadal count of off-season TCs has been found to be strongly associated with climate change trends in sea surface temperature (SST), global mean surface temperature (GMST) and cloud cover (CC) in all three basins (EP, NA and WP). The MLR model where climate variability variables (ENSO and AMO) explained most of the variance in off-season TC counts was in the storm decadal counts for the post-season months of the NA basin.

Results of the MLR stepwise procedures showed that the selected variables that accounted for most of the variability in off-season TCs for the EP basin were SST, CC and ENSO, while CC (pre-season and off-season) and AMO (post-season) were chosen as the best variables for the NA basin. The stepwise procedure identified the climate change trends in GMST and CC, and fluctuations in ENSO and IPO as the variables that accounted for most of the variability in decadal off-season total TC counts in the WP basin,

The findings of this study suggest that trends in SST, GMST and CC associated with climate change are not only altering the frequency (Klotzback, 2006; Saunders and Lea, 2006; Hansen et al., 2010)and intensity of TCs that develop during the peak months of the season, they are also altering the total number of storms that form in the off-season months (Dec-May), especially in the EP and NA basins. The results of this study have important implications for the NA and EP basins, if off-season TCs have been increasing in frequency since the 1900 we can expect that this trend associated with climate change would continue in future decades. This increasing number of off-season TCs could potentially impact societies in their path during times of the year when storms are least expected, possibly increasing environmental and economic impacts in areas that are already experiencing the effects of climate change exacerbated phenomena.

**Acknowledgments**

To Roberto García, director of the National Meteorological Service of San Juan, Puerto Rico, for his suggestions during the discussion of this paper. The 6- hourly tropical cyclone track data supporting this article are based on publicly available measurements from the International Best Track Archive for Climate Stewardship (IBTrACS; https://www.ncdc.noaa.gov/ibtracs/). The sea surface temperature (HadISST1 1° reconstruction), and cloud cover (ICOADS v2.5 1°) datasets supporting this article are based on publicly available measurements that can be accessed from the Kingdom of the Netherlands Meteorology Institute (KMNI; https://climexp.knmi.nl/start.cgi). The global mean surface temperature (GISTEMP v4) data supporting this article are based on publicly available measurements that from the NASA Goddard Institute for Space Studies (GISS; https://climatedataguide.ucar.edu/climate-data/global-surface-temperature-data-gistemp-nasa-goddard-institute-space-studies-giss.) The El Niño Southern Oscillation (ENSO 3.4) data supporting this article are based on publicly available measurements from the National Oceanic and Atmospheric Administration Physical Sciences Lab (PSL; https://psl.noaa.gov/gcos_wgsp/Timeseries/Data/nino34.long.data). The Atlantic Multidecadal Oscillation data supporting this article are based on publicly available measurements from the National Center for Atmospheric Research (NCAR; https://climatedataguide.ucar.edu/climate-data/atlantic-multi-decadal-oscillation-amo). The Interdecadal Pacific Oscillation data supporting this article are based on publicly available measurements from the National Oceanic and Atmospheric Administration Physical Sciences Lab (PSL; https://psl.noaa.gov/data/timeseries/IPOTPI/).





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

Index for the Interdecadal Pacific Oscillation. Climate Dynamics, 45(11-12), 3077-
3090.http://dx.doi.org/10.1007/s00382-015-2525-1 . Accessed on MM DD YYYY at
"/data/timeseries/IPOTPI".
Hipel, K.W. and McLeod, A.I., (2005). Time Series Modelling of Water Resources and
Environmental Systems. Electronic reprint of our book originally published in 1994.
http://www. stats.uwo.ca/faculty/aim/1994Book/
Holland, G.J. and P.J. Webster (2007) Heightened tropical cyclone activity in the North Atlantic:
Natural variability or climate trend? *Philos. Trans. Roy. Soc.*, 365: 2695–2716.
https://doi.org/10.1098/rsta.2007.2083
James, Gareth, Daniela Witten, Trevor Hastie, and Robert Tibshirani. 2014. An Introduction to
Statistical Learning: With Applications in R. Springer Publishing Company, Incorporated.
Knapp, K. R., H. J. Diamond, J. P. Kossin, M. C. Kruk, C. J. Schreck, 2018:International Best
Track Archive for Climate Stewardship (IBTrACS) Project, Version 4. [indicate subset used].
NOAA National Centers for Environmental Information. non-gonvernment domain
https://doi.org/10.25921/82ty-9e16
Knutson, T.R., J.L. McBride, J. Chan, K. Emanuel, G. Holland, C. Landsea, I. Held, J.P. Kossin,
A.K. Srivastava and M. Sugi (2010) Tropical cyclones and climate change. *Nature Geoscience,*
**3**: 157–163; doi:10.1038/ngeo779
Klotzbach, P.J. (2006) Trends in global tropical cyclone activity over the past twenty years (1986–
2005)  Geophys. Res. Lett., 33: L10805, pp 1-4. https://doi.org/10.1029/2006GL025881
Landsea, C.W. (2005) Meteorology: Hurricanes and global warming. *Nature*, 438: E11–E12.
https://doi.org/10.1038/nature04477



Landsea, C.W., B.A. Harper, K. Hoarau, J.A. Knaff (2006) Can we detect trends in extreme tropical
cyclones? *Science 2006,* **313**:452–454. DOI: 10.1126/science.1128448
Landsea, C. (2007). Counting Atlantic tropical cyclones back to 1900. Eos, Transactions American
Geophysical Union, 88(18), 197-202. https://doi.org/10.1029/2007EO180001
Mahala, B. K., Nayak, B. K., & Mohanty, P. K. (2015). Impacts of ENSO and IOD on tropical
cyclone activity in the Bay of Bengal. *Natural Hazards*, *75*(2), 1105-1125.
https://doi.org/10.1007/s11069-014-1360-8
Mann, H.B. (1945), Nonparametric tests against trend, Econometrica, 13, 245-259.
Mann, M.E., T.A. Sabbatelli and U. Neu (2007) Evidence for a modest undercount bias in early
historical Atlantic tropical cyclone counts. Geophys. Res. Lett., 34: L22707;
doi:10.1029/2007GL031781
Mann, M.E., J.D. Woodruff, J. Donnelly and Z. Zhihua (2009) Atlantic hurricanes and climate over
the past 1,500 years. Vol. 460 (13 August 2009); doi:10.1038/nature0821.
McLeod, A. I. (2005). Kendall rank correlation and Mann-Kendall trend test. R Package Kendall.
Nogueira, R. C., Keim, B. D., Brown, D. P., & Robbins, K. D. (2013). Variability of rainfall from
tropical cyclones in the eastern USA and its association to the AMO and ENSO. *Theoretical*
*and applied climatology*, *112*(1-2), 273-283. https://doi.org/10.1007/s00704-012-0722-y
Saunders, M., Lea, A. Large contribution of sea surface warming to recent increase in Atlantic
hurricane activity. *Nature* 451, 557–560 (2008) doi:10.1038/nature06422
Trenberth, K.E. and D.J. Shea (2006) Atlantic hurricanes and natural variability in 2005. *Geophys.*
*Res. Lett.,* 33: L12704, doi:10.1029/2006GL026894
Trenberth, K. E. (2007). Warmer oceans, stronger hurricanes. Scientific American, 297(1), 44-51.
https://www.jstor.org/stable/26069374
Trenberth, Kevin, Zhang, Rong & National Center for Atmospheric Research Staff (Eds). Last
modified 10 Jan 2019. "The Climate Data Guide: Atlantic Multi-decadal Oscillation (AMO)."
Retrieved from https://climatedataguide.ucar.edu/climate-data/atlantic-multi-decadal-
oscillation-amo.
Venables, W. N. and Ripley, B. D. (2002) Modern Applied Statistics with S. Fourth edition.
Springer.
Walsh, K. J., McBride, J. L., Klotzbach, P. J., Balachandran, S., Camargo, S. J., Holland, G., ... &
Sugi, M. (2016). Tropical cyclones and climate change. *Wiley Interdisciplinary Reviews:*
*Climate Change*, *7*(1), 65-89.L doi: 10.1002/wcc.371
Wang, C., & Lee, S. K. (2008). Global warming and United States landfalling hurricanes.
Geophysical Research Letters, 35(2). https://doi.org/10.1029/2007GL032396
Zhao, J., Zhan, R., Wang, Y., & Xu, H. (2018). Contribution of the interdecadal Pacific oscillation
to the recent abrupt decrease in tropical cyclone genesis frequency over the western North
Pacific since 1998. Journal of Climate, 31(20), 8211-8224. https://doi.org/10.1175/JCLI-D-18-
539     0202.1