# Peer review of "Increasing Frequency in Off-Season Tropical Cyclones and its relation to Climate Variability and Change"

_Weather and Climate Dynamics, 2020_

## Referee Comment (RC1) · Anonymous Referee #1 · 9 Sep 2020

José J. Hernández Ayala and
Rafael Méndez-Tejeda

**Anonymous Referee #1**

This study examines the decadal change in the TC frequency during TC-inactive seasons and its linkage to climate variability. It is well written and shows a few interesting results. However, the results of this study are primarily based on the unreliable and unconsistent TC best tracks. It is not known whether the findings of this study are induced by artificial effect or real physics. Therefore, I suggest a major revision.

Specific comments: 1. The authors have already mentioned that TC best tracks before the satellite era is unreliable. Regardless of what modification that they apply, the modified data is still of lower quality. There are several decades from 1966 to present, which

is long enough for an analysis on the decadal time scale. Thus I would recommend the authors analyze the data since 1966.

2. Another issue related to TC data is the uncertainty in observing the weakest TCs, e.g. tropical depressions. The observation of tropical depressions is highly sensitive to the TC-detecting technologies. I would suggest the author exclude tropical depressions in a revised manuscript.

3. Since the MK test is a well-documented method to detect potential trends, there is no need to represent the detailed algorithm in the paper.

4. Where is the cloud cover dataset obtained from? Before the introduction of satellites, are these cloud cover data reliable?

5. Considering the increasing TC frequency shown by the authors and global warming, it is natural that the correlation between TC frequency and GMST is significant. To make this point more clear, the author should further examine the spatial patterns of the changes in the TC occurrence and the SST. Does the region with rising SST correspond to the region with more TC formation?

6. Since the significant increasing trend in the TC frequency, the correlation between TC number and climate indices might be reduced. Therefore, I would suggest the authors compute the correlation coefficients after removing long-term trends, to highlight the potential relationship on the decadal time scale.

---

## Referee Comment (RC2) · Jhordanne Jones (Referee) · 22 Sep 2020

**Overview:**
Hernandez and Mendez-Tejeda assess trends in off-season storms in global hurricane/typhoon basins and attempts to associate these changes to warming sea surface temperatures and both local and remote climate phenomena. Storms occurring outside the dominant peak are not a common topic of study and are certainly worth looking into. The manuscript is well-written and well-organized, and its results may have implications for global awareness of changes to "off-season" storm activity with

climate change. I have outlined some comments and questions to the authors below. My comments outline some suggestions and considerations that could help make the manuscript a bit clearer to the reader. The methods are scientifically sound, and the conclusions are reasonable. But I'd also caution the authors on making conclusions about trends in the frequency of off-season storms without also assessing the frequency of in-season storms as well and not so much rely on the conclusions of previous studies. While there is no significant trend in global storm frequency, the significance may vary with individual basins. My recommendation is for a major revision.

**Major Comments:**

1. Do not confuse the official storm season with storm climatology. While it is true that there are set dates outlining the official hurricane/typhoon seasons, I think it should also be mentioned that this does not mean that it is atypical for storms to occur outside the official season. Storms occur where environmental conditions are relatively favorable. For the Atlantic, the climatological environment becomes favorable in the spring, which means storms are possible as early as April or May and as late as December. However, they usually do not compare to in-season storms in terms of overall numbers, intensity and duration. This is perhaps why May and December is not counted in the Atlantic's official season despite it being very possible to have storms (albeit not very strong ones) develop within these months.

2. I understand it can be pretty tricky to define what "off-season" means. But in my opinion, given the length of each basin's climatological season, there isn't much sense in categorizing storms into pre- versus post-season storms. Just looking at Figures 1(b-e), there are only about 2-4 months out of the year in which the environment is drastically unfavorable for each basin. But, this is a comment and

not really a suggestion to change your methodology at this point.

3. I think a good way to assess whether there is any meaningful change in the number of off-season storms is to show changes in the climatology over time. Just showing the number of off-season storms outside the context of in-season storms makes it harder to interpret the trend. The number of off-season storms is likely related to what happens during the official season since they are driven by the same climate variability and change (as suggested by the papers referenced).

4. I agree with Referee 1 that the manuscript lacks a physical explanation or mechanism for the climate associations the authors make. In what way do SST, GMST and CC alter the number of off-season storms? Is it a shift in the climatology of storms or is there also an increase of storms within the official seasons? How have climate dynamics changed over time in months prior to/post the official season to cause the observed trend without also causing any significant trend in in-season storm frequency? This could help augment your discussion section and provide a better comparison to previous studies.

**Minor Comments:**

1. Please go through the manuscript carefully to correct grammatical errors, unfinished sentences and repetition.

2. The authors do not define the regions over which sea surface temperature and cloud cover are averaged in each basin. Similarly, how are the ENSO, AMO, IOD and IPO indices defined and over what region?

3. You mention statistical significance quite a bit. But, at what value of Kendall's tau is statistical significance achieved and what frequency is this associated with for each basin?

4. Line 49: Walsh et al. (2016) could be updated to the more recent Walsh et al. (2019). There is also Knutson et al. (2019, 2020) for an updated review of tropical cyclone trends and projections.

   (a) Knutson, T., and Coauthors, 2019: Tropical Cyclones and Climate Change Assessment: Part I: Detection and Attribution. Bull. Amer. Meteor. Soc., 100, 1987–2007, https://doi.org/10.1175/BAMS-D-18-0189.1.

   (b) Knutson, T., and Coauthors, 2020: Tropical Cyclones and Climate Change Assessment: Part II: Projected Response to Anthropogenic Warming. Bull. Amer. Meteor. Soc., 101, E303–E322, https://doi.org/10.1175/BAMS-D-18-0194.1.

   (c) Walsh, K. J., Camargo, S. J., Knutson, T. R., Kossin, J., Lee, T. C., Murakami, H., Patricola, C., 2019: Tropical cyclones and climate change. Tropical Cyclone Research and Review, 8(4), 240-250, https://www.sciencedirect.com/science/article/pii/S2225603220300047.

5. Lines 73-76: Is there a reference or citation for this sentence? Yes, the decade does have the most off-season storms but this could be due to more recent hurricane/typhoon seasons being better observed and measured. And again, May has been the climatological start of most seasons.

6. I think the details outlined in Sections 3 and 4 can be summarized. Further details can be placed in an appendix or supplementary document. Or you can just direct the reader to the proper citations.

7. The stacked columns used in the figures can be incredibly confusing. It's hard to infer any conclusion about trends in pre versus post-season numbers.

8. Is there a baseline number of off-season storms in each basin to compare changes to? How do you assess a statistically significant change or trend?

9. I really like that the p-value column was included in Tables 3 and 4. Not many studies do this.

10. Figure 1(a) doesn't really help since the darker tracks hide the lighter tracks, particularly for the Western North Pacific. A good alternative would be to replace 1(a) with a figure of storm climatology (as you did in Figures 1[b-e]) over the five categories you've created. This would probably address my Major Comment 3 above and most of my comments on changes in trend. You should also note in the caption that Figures 1 (b-e) all have different scales of frequency.

11. In Figures 3-5, all you do is superpose the trends in GMST, ENSO and IPO/AMO onto the same figure of pre and post-season storm numbers. What if you regressed total off-season storm frequency against each decadal index to show how each climate phenomena influenced storm numbers over time?

Please also note the supplement to this comment:
https://wcd.copernicus.org/preprints/wcd-2020-36/wcd-2020-36-RC2-supplement.pdf

---

## Author Comment (AC1) · 19 Oct 2020

This study examines the decadal change in the TC frequency during TC-inactive seasons and its linkage to climate variability. It is well written and shows a few interesting results. However, the results of this study are primarily based on the unreliable and inconsistent TC best tracks. It is not known whether the findings of this study are induced by artificial effect or real physics. Therefore, I suggest a major revision.

[Figure]

Specific comments:

1. The authors have already mentioned that TC best tracks before the satellite era is unreliable. Regardless of what modification that they apply, the modified data is still of lower quality. There are several decades from 1966 to the present, which is long enough for an analysis on the decadal time scale. Thus, I would recommend the authors analyze the data since 1966.

The authors appreciate the referee's comments on the temporal limitations of the TC best tracks dataset. However, many studies have used the pre-satellite era TC track data to examine trends in TC frequency (Landsea et al. 2006; Landsea 2007; Mann et al, 2007; Mann et al, 2009), of course applying corrections for the TCs that might have stayed over the ocean. We are aware that the modifications done to the pre-satellite best track data is of lower quality than the post-satellite data, yet we argue that even with those limitations associated with the track count corrections, our results should be presented for both the entire period starting in the 1900s and for 1966 to present. In the paper we also discussed the results since 1966, this is what we wrote:

"The EP and NA basins exhibited statistically significant increasing trends even if the analysis was done from the 1960s instead of the 1900s. The WP basin showed an overall increasing trend in the total number of off-season TCs per decade, yet if the analysis is done from the 1960s to the present, no statistically significant increasing trend is found. However, the three basins that reflected an overall increase in decadal off-season TC frequency had their most active decades after the 1970s."

2. Another issue related to TC data is the uncertainty in observing the weakest TCs, e.g. tropical depressions. The observation of tropical depressions is highly sensitive to TC-detecting technologies. I would suggest the author exclude tropical depressions in a revised manuscript.

We understand the reviewer's concern with the uncertainty in observing the weakest TCs, e.g. tropical depressions. However, a tropical depression is still a tropical cyclone

and we believe that they should be included in any kind of analysis of TC frequency. Since our study focuses on off-season storms, we are already working with a limited number of TCs, and excluding the weaker ones might be detrimental to our analysis that focuses on the frequency and not on TC intensity. We will be examining the off-season TC intensity question in future research.

3. Since the MK test is a well-documented method to detect potential trends, there is no need to represent the detailed algorithm in the paper.

The authors agree with the reviewer's suggestion of removing the MK test equations from the paper.

4. Where is the cloud cover dataset obtained from? Before the introduction of satellites, are these cloud cover data reliable?

The Cloud Cover dataset used in this study comes from the International Comprehensive Ocean-Atmosphere Data Set (ICOADS) which offers surface marine data spanning the past three centuries, and simple gridded monthly summary products for $2°$ latitude x $2°$ longitude boxes back to 1800 (and $1°$x$1°$ boxes since 1960)—these data and products are freely distributed worldwide. I got this from their website "As it contains observations from many different observing systems encompassing the evolution of measurement technology over hundreds of years, ICOADS is probably the most complete and heterogeneous collection of surface marine data in existence." Similar to the SST used in this study, it seems that the CC data from ICOADS is also reliable. Check this for more info: https://icoads.noaa.gov/icoads_brochure_20160308_8.5x11.pdf

5. Considering the increasing TC frequency shown by the authors and global warming, it is natural that the correlation between TC frequency and GMST is significant. To make this point clearer, the author should further examine the spatial patterns of the changes in the TC occurrence and the SST. Does the region with rising SST? correspond to the region with more TC formation?

The authors appreciate the referee's suggestion to examine the spatial relationship between rising SST and off-season TC formation. For that reason, we decided to add the off-season TC tracks to Figures 3, 4, and 5 and there we show that off-season TC occurrence has been in areas that have experienced statistically significant increasing trends in SST and CC. On the last page of this document, you find an example of the updated figure (Fig 1).

6. Since the significant increasing trend in the TC frequency, the correlation between TC number and climate indices might be reduced. Therefore, I would suggest the authors compute the correlation coefficients after removing long-term trends, to highlight the potential relationship on the decadal time scale.

We removed the long-term trends for the basins that showed statistically significant trends and for the SST time series and we found that the correlations were still significant even after removing the trend using the constant method (e.g., the mean value over the entire series) to detrend. Here are the results for the off-season TCs in the Atlantic and SST after detrending both series. In Fig.2 of this response, we show a graph of the detrended series.

Coefficients: Estimate Std. Error t value Pr(>|t|) (Intercept) -284.651 95.388 -2.984 0.0137 * Detrend_Data$'NA 10.924 3.587 3.046 0.0123 * — Signif. codes: 0 '***' 0.001 '**' 0.01 '*' 0.05 '.' 0.1 ' ' 1

Residual standard error: 2.783 on 10 degrees of freedom Multiple R-squared: 0.4812, Adjusted R-squared: 0.4293 F-statistic: 9.276 on 1 and 10 DF, p-value: 0.01234

Please also note the supplement to this comment:
https://wcd.copernicus.org/preprints/wcd-2020-36/wcd-2020-36-AC1-supplement.pdf

[Figure]

[Figure]

**Fig. 1.** Decadal TC counts for the NA off-seasons and decadal average SSTs (a), decadal TC counts for the NA off-seasons and decadal average AMO (b), decadal TC counts for the NA off-seasons and decadal averag

[Figure]

**Fig. 2.** Detrended TC and SST decadal series for the North Atlantic

---

## Author Comment (AC2) · 19 Oct 2020

Major Comments:

1. Do not confuse the official storm season with storm climatology. While it is true that there are set dates outlining the official hurricane/typhoon seasons, I think it should also be mentioned that this does not mean that it is atypical for storms to occur outside the official season. Storms occur where environmental conditions are relatively favorable. For the Atlantic, the climatological environment becomes favorable in the spring, which

means storms are possible as early as April or May and as late as December. However, they usually do not compare to in-season storms in terms of overall numbers, intensity and duration. This is perhaps why May and December is not counted in the Atlantic's official season despite it being very possible to have storms (albeit not very strong ones) develop within these months.

The authors agree with the reviewers comments and we added a sentence in the last paragraph of the introduction the mentions that is not atypical for TCs to develop outside of the core season months of each basin. In this manuscript we focus on the months with the lowest frequency of TCs and not necessarily on the official TC season for each basin, yet we think that making that clarification is important so thank you for the suggestion.

2. I understand it can be pretty tricky to define what "off-season" means. But in my opinion, given the length of each basin's climatological season, there is not much sense in categorizing storms into pre- versus post-season storms. Just looking at Figures 1(b-e), there are only about 2-4 months out of the year in which the environment is drastically unfavorable for each basin. But this is a comment and not really a suggestion to change your methodology at this point.

The authors understand the reviewer's concern, yet we believe that by examining the pre and post-seasons TC frequencies and trends we are able to focus on the temporal particularities of each basin and whether there are more/less storms developing over time in the pre/post-seasons. In the manuscript we also analyzed all of the off-season TCs together without separating them into pre and post-seasons, so that addresses some of the reviewer's concerns.

3. I think a good way to assess whether there is any meaningful change in the number of off-season storms is to show changes in the climatology over time. Just showing the number of off-season storms outside the context of in-season storms makes it harder to interpret the trend. The number of off-season storms is likely related to what

happens during the official season since they are driven by the same climate variability and change (as suggested by the papers referenced).

The authors understand the reviewer's concerns, yet we would like to keep our focus on the off-season TC climatology since the in-season TCs have been studied by many researchers already and no clear increasing trend has been found.

4. I agree with Referee 1 that the manuscript lacks a physical explanation or mechanism for the climate associations the authors make. In what way do SST, GMST and CC alter the number of off-season storms? Is it a shift in the climatology of? storms or is there also an increase of storms within the official seasons? How have climate dynamics changed over time in months prior to/post the official season to cause the observed trend without also causing any significant trend in in-season storm frequency? This could help augment your discussion section and provide a better comparison to previous studies.

Our hypothesis is the following, since climate change is causing an increase in SST (which also leads to an increasing trend in CC) in areas where TCs develop, we wanted to see if those increasing trends that are also occurring in the low activity months could be behind the increasing number of storms developing in those months where few TCs form. Again, we think that a lot of researchers have already studied the relationship between TC frequency during the peak season months of activity and climate variability and change, and for that reason we think is necessary to focus on the off-season TCs since we don't know if climate variability or change are causing changes in their frequency.

Minor Comments:

1. Please go through the manuscript carefully to correct grammatical errors, unfinished sentences and repetition.

We read the manuscript thoroughly and grammatical errors, unfinished sentences and

repetitions were fixed.

2. The authors do not define the regions over which sea surface temperature and cloud cover are averaged in each basin. Similarly, how are the ENSO, AMO, IOD and IPO indices defined and over what region?

Thank you for pointing out that missed information, the SST and CC were averaged over the 5-25° degrees north in the Atlantic and Pacific Basins. The ENSO 3.4, AMO and IPO indexes were all averaged from their respective areas of development.

3. You mention statistical significance quite a bit. But, at what value of Kendall's tau is statistical significance achieved, and what frequency is this associated with for each basin?

Based on our results, the lowest Kendall's Tau value at which statistical significance is achieved is 0.485. I think we made this clear in Table 2.

Table 2. Results of Mann-Kendall trend tests for the 1900-2019 period for all ocean basins where TCs form.

Pre TCs Tau S P-value Post TCs Tau S P-value Off TCs Tau S P-value EP 0.746 0.002 EP 0.098 0.723 EP 0.679 0.004 NA 0.572 0.015 NA 0.485 0.042 NA 0.554 0.016 SP 0.048 0.889 SP 0.015 1.000 SP 0.061 0.836 WP 0.554 0.016 WP 0.485 0.034 WP 0.534 0.019 Significant trends in bold.

4. Line 49: Walsh et al. (2016) could be updated to the more recent Walsh et al. (2019). There is also Knutson et al. (2019, 2020) for an updated review of tropical cyclone trends and projections.

Thank you for the suggestion, the papers have been updated to the most recent versions.

(a) Knutson, T., and Coauthors, 2019: Tropical Cyclones and Climate Change Assessment: Part I: Detection and Attribution. Bull. Amer. Meteor. Soc., 100, 1987–2007,

https://doi.org/10.1175/BAMS-D-18-0189.1.

(b) Knutson, T., and Coauthors, 2020: Tropical Cyclones and Climate Change Assessment: Part II: Projected Response to Anthropogenic Warming. Bull. Amer. Meteor. Soc., 101, E303–E322, https://doi.org/10.1175/ BAMS-D-18-0194.1.

(c) Walsh, K. J., Camargo, S. J., Knutson, T. R., Kossin, J., Lee, T. C., Murakami, H., Patricola, C., 2019: Tropical cyclones and climate change. Tropical Cyclone Research and Review, 8(4), 240-250, https://www.sciencedirect.com/science/article/pii/S2225603220300047.

5. Lines 73-76: Is there a reference or citation for this sentence? Yes, the decade does have the most off-season storms, but this could be due to more recent hurricane/ typhoon seasons being better observed and measured. And again, May has been the climatological start of most seasons.

You are correct in pointing out that this is due to more recent hurricane/ typhoon seasons being better observed and measured, yet it still important to mentioned since it was still the season that produced the most off-season storms in the satellite era.

6. I think the details outlined in Sections 3 and 4 can be summarized. Further details can be placed in an appendix or supplementary document. Or you can just direct the reader to the proper citations.

Sections 3 and 4 have been summarized and all of the algorithms associated with the MK Test have been removed.

7. The stacked columns used in the figures can be incredibly confusing. It is hard to infer any conclusion about trends in pre versus post-season numbers.

We understand the reviewer's concern with figure 2, however, we think is important to show the total counts by decade for the pre and post off-season TCs.

8. Is there a baseline number of off-season storms in each basin to compare changes

to? How do you assess a statistically significant change or trend?

Her we selected all of the TCs that developed outside of the peak-season months for each basin and we summed them all by decades. Those decadal counts were the ones that were analyzed with the MK test for trends.

9. I really like that the p-value column was included in Tables 3 and 4. Not many studies do this.

Thank you for letting us know!

10. Figure 1(a) does not really help since the darker tracks hide the lighter tracks, particularly for the Western North Pacific. A good alternative would be to replace 1(a) with a figure of storm climatology (as you did in Figures 1[b-e]) over the five categories you have created. This would probably address my Major Comment 3 above and most of my comments on changes in trend. You should also note in the caption that Figures 1 (b-e) all have different scales of frequency.

We appreciate the reviewer's suggestion with regards to figure 1, yet we think is important to keep that figure since it shows all of the off-season TCs analyzed in this study. We did modified figures 3, 4, and 5 to show that most off-season TCs are forming in the areas that have been experiencing an increasing trend in SSTs. We also fixed the different scales of frequency in Fig 1, thank you for letting us know about that.

11. In Figures 3-5, all you do is superpose the trends in GMST, ENSO, and IPO/AMO onto the same figure of pre-and post-season storm numbers. What if you regressed total off-season storm frequency against each decadal index to show? how each climate phenomenon influenced storm numbers over time?

We actually did that in the study, and we discuss the results of the multiple linear regression and stepwise procedures in Tables 3 and 4. We regressed the climate change and variability series with pre, post, and total off-season TC counts and we discussed the results. In the attached supplement file you will see the tables where

the results of those analyses are presented.

Please also note the supplement to this comment:
https://wcd.copernicus.org/preprints/wcd-2020-36/wcd-2020-36-AC2-supplement.pdf

---

## Author Response (AR3)

Comments to the Author:

I had some follow-up discussions about this paper with the two reviewers and we decided to ask you for a final round of revisions to further strengthen the quality of the paper. Please find the main points we would like you to address below. We will try to look into this quickly after you return the revised manuscript in order to avoid further delays in the publication process.

1) In-season trend: While it is not necessary to analyze the in-season trend in detail yourself, you should expand the discussion on the relationship between in- and off-season trends. State clearly what we know from the literature about long-term trends / decadal variability / relation to climate indices for in-season / total storm counts. Then discuss your own findings in the light of these literature results. Give possible reasons why off- and in-season counts may differ. If you think that global warming affects the off-season (and thus more marginal) storms in a disproportional way, you need to clearly state this as a hypothesis you are testing in this paper.

Thank you for the suggestion, we added this paragraph at the end of the results and discussion section:

**"The results of the MLR and stepwise procedures have shown that increasing trends in decadal off-season TCs in the NA, EP and WP basins are mostly associated with climate change (SST and CC) and climate variability factors (ENSO and IPO). Since no previous studies have focused on analyzing trends in off-season TC trends, our results can only be compared to analyses that have consider in-season storms. Studies that have examined TC frequency overall have found increases in the number of most intense hurricanes [Wang and Lee, 2008; Knutson et al., 2010; Emanuel, 2013], yet no clear trend has been found when lower intensity TCs have been examined [Landsea, 2007]. The results of other studies show that there is no overall agreement on the relationship between SSTs and TC frequency (Landsea, 2005; Emanuel, 2005; Trenberth and Shea, 2006; Trenberth, 2007), yet some have found strong associations between TC variability and ENSO, AMO and IPO [Camargo and Sobel, 2005; Nogueira and Kim, 2007; Mahala et al., 2015; Zhao et al., 2018]. In this study we analyzed all off-season TCs and our results differ from those that have found no trend in overall TC frequency, since we found decadal increasing trends in the NA and EP basin in both the pre and post-satellite eras. It is important to note that the findings of this study are different from other analyses since here the focus is on analyzing trends in off-season TCs, which has not been done before. The results presented here suggest that climate change trends like higher SSTs and more favorable moisture environments (CC) between the months of Dec to May in the NA and EP basins seem to be the major factors behind decadal increasing trends in off-season TCs."**

2) Trends in climate indices: Clearly explain the differences in your MLR models with the full and detrended climate indices. What is the role of the strong trends we see in Figs. 3–5? Why is the ENSO index below zero most of the time in Figs. 3b and 5b?

Thanks for the suggestion, we have added the results of the MLR models with the full and detrended climate indices in Table 3, check table updated table below. The role of the strong trends in Fig 3-5 suggests that climate change trends are the dominant factor between the increasing trend of off-season TCs in the NA and EP basins. The ENSO index is mostly below zero because it is a decadal average for the ENSO 3.4 index.

Table 3. Best multiple linear regression models (MLR) for basins with statistically significant increasing trends in off-season TCs with detrended climate indices.

| Model | Adj. R² | **Adj. R² Det.** | Factors | p-val. | **p-val. Det.** |
|---|---|---|---|---|---|
| EP pre-season | 0.563 | **0.444** | SST, GMST & CC | 0.021 | **0.038** |
| EP off-season | 0.522 | **0.472** | SST, GMST & CC | 0.030 | **0.024** |
| NA pre-season | 0.481 | **0.496** | SST, GMST & CC | 0.041 | **0.022** |
| NA post-season | 0.427 | **0.247** | ENSO & AMO | 0.130 | **0.057** |
| NA off-season | 0.384 | **0.406** | SST, GMST & CC | 0.070 | **0.010** |
| WP pre-season | 0.551 | **0.462** | SST, GMST & CC | 0.020 | **0.000** |
| WP post-season | 0.645 | **0.478** | SST, GMST & CC | 0.000 | **0.023** |
| WP off-season | 0.689 | **0.481** | SST, GMST & CC | 0.005 | **0.017** |

3) Pre-satellite era: Be more explicit about the differences in trends over the full period and those since the 1960s. Give numbers and significance levels.

Table 2 has been updated and it now includes MK trends test results for the post-satellite era decades. However, it seems that for the MK test to work properly you need at least 10 observations to analyze over time and when the test is done for the decades from 1960 to the present it only has 6 observations (6 decades). That is probably the reason why the Tau coefficients for the pre-season TCs are all the same. The results for the pos-satellite data show no significant statistical trends. Here is the modified table,

**Table 2. Results of Mann-Kendall trend tests for the 1900-2019 period for all ocean basins where TCs form.**

| Pre TCs | Tau S | P-value | Post TCs | Tau S | P-value | Off TCs | Tau S | P-value |
|---|---|---|---|---|---|---|---|---|
| | | | Trends for the 1900-2019 period | | | | | |
| EP | **0.746** | **0.002** | EP | 0.098 | 0.723 | EP | **0.679** | **0.004** |
| NA | **0.572** | **0.015** | NA | **0.485** | **0.042** | NA | **0.554** | **0.016** |
| SP | 0.048 | 0.889 | SP | 0.015 | 1.000 | SP | 0.061 | 0.836 |
| WP | **0.554** | **0.016** | WP | **0.485** | **0.034** | WP | **0.534** | **0.019** |
| | | | Trends for the 1960-2019 period | | | | | |
| EP | 0.596 | 0.158 | EP | 0.414 | 0.338 | EP | 0.69 | 0.085 |
| NA | 0.596 | 0.158 | NA | 0.645 | 0.119 | NA | 0.6 | 0.132 |
| SP | 0.596 | 0.158 | SP | -0.2 | 0.707 | SP | -0.06 | 1 |
| WP | -0.467 | 0.259 | WP | -0.6 | 0.132 | WP | -0.69 | 0.085 |

**Significant trends in bold.**

4) Data homogenization: Better discuss the modifications applied to the data in the pre-satellite era. There is literature about this for the North Atlantic (Landsea et al. 2006; Landsea 2007; Mann et al,

2007; Mann et al, 2009) but not for the other basins. Critically discuss the situation for the other basins. What evidence do we have that modifications are reliable?

We are aware that the modification to the pre-satellite era done here has not been applied elsewhere, yet we think is a method that could produce similar results to those found in the NA basin since the Pacific basins where TCs develop have been well observed. We think that the method used by Landsea 2007 is useful for all basins since it is based on a very basic assumption, that the TC landfall percentage of the pre-satellite era should be similar to the landfall percentage of the post-satellite era. Of course, the method is not perfect, there might be some TCs that were not detected by this method or by actual observations. The only basin that might be a little bit more complicated is the EP, where fewer TCs make landfall. We added this sentence to section 4 of the paper to clarify the limitations of this method:

"It is important to note that this TC adjustment method has been only implemented in the NA basin and that this study is the first attempt to apply this missed storm adjustment technique in other ocean basins. This adjustment method is in no way capable of detecting all TCs that formed before the satellite era, yet it offers us the opportunity to estimate missed storms by comparing the TC landfall percentage of the pre- and post-satellite eras."

5) Tropical depressions: Moon et al. (2019) report that there is great uncertainty in detecting tropical depressions, even in the satellite era. Recompute your results taking out tropical depressions to check whether your conclusions are robust.

When all of the TCs with unknown intensities are removed, since only a few of them have actual wind speeds, we cut our total TC numbers by more than half from 743 to 342 for the NA, EP and WP basins. The issue with removing those NR (intensity not reported) TCs is that we might also be removing storms that were stronger than tropical depressions. Since we do not know the actual intensities of those storms, we think is irresponsible to assume that they are all tropical depressions. We understand the reviewers concerns with adding TDs to the analysis, yet we think that all TCs that have been observed should be included in the analysis. For that reason, we don't think is responsible to compute our results taking out tropical depressions since we would also run the risk of eliminating storms that could have been tropical storms of higher intensities.

6) TC tracks and SST change: Better explain the relationship between SST trends and changes in TCs in Figs. 3–5. Were there any changes over time in track or genesis location?

Thank you for the suggestion, yet we believe that the figures show that most TCs formed and developed over areas that exhibited statistically significant increasing trends in SST.

7) Relationship to cloud cover: You show a significant correlation but should give a better explanation of a potential physical link.

Thank you for the suggestion. We have added these sentences explaining the physical links in previous manuscripts:

**"Increasing trends in SSTs, GMST and moisture (CC) outside of the prime months of tropical storm development could promote more optimum conditions for higher off-season TC occurrences (Klozback, 2006; Hansen et al., 2010)."**

**Overall, these results suggest that increasing trends in SSTs, which also drive increasing trends in evaporation rates associated with high CCs, are the physical mechanisms behind most of the recent increase in the total number of out of season TCs in the NA basin.**

We also added this sentence to the final paragraph of the results and discussion section

**"The results presented here suggest that climate change trends like increasing SSTs and more favorable moisture environments (CC) between the months of Dec to May in the NA and EP basins seem to be the major factors behind decadal increasing trends in off-season TCs."**

8) ICOADS: Add more detail on the limitations of ICOADS, citing relevant literature (Freeman et al. 2016, Eastman et al. 2011). Can you justify the use of ICOADS cloud cover data before the 1950s?

The citations have been added to the manuscript, thank you for sharing them. I think this sentence in Freeman et al. 2016 can give us a reason to justify the use of ICOADS data before the 1950s;

"Figure **4** shows the coverage of reports containing total cloud cover ($N$) needed to estimate solar radiation, and in the evaporation parameter ($G$) shows the coverage for those reports containing variables needed to estimate either latent or sensible heat flux from mean parameters (Berry and Kent, **2009**). There are noticeable increases in ocean coverage for clouds in the 1850s and 1860s and around 1900. The surface flux coverage illustrated by $G$ is reduced by the lack of availability of humidity measurements in many reports with consistent coverage only from the 1910s."

[revised manuscript text omitted]

---

## Author Response (AR4)

Response to the Co-Editor #3

1) In-season trend: The text you added is sufficient to put your results into the context of other studies. However, there are some redundancies in this paragraph, and you should try to streamline it a bit.

Thank you for the suggestion. We have reduced the paragraph by removing the first two sentences and one of the sentences at the end that were redundant, and it now reads like this:

**"Studies that have examined TC frequency overall have found increases in the number of most intense hurricanes [Wang and Lee, 2008; Knutson et al., 2010; Emanuel, 2013], yet no clear trend has been found when lower intensity TCs have been examined [Landsea, 2007]. The results of other studies show that there is no overall agreement on the relationship between SSTs and TC frequency (Landsea, 2005; Emanuel, 2005; Trenberth and Shea, 2006; Trenberth, 2007), yet some have found strong associations between TC variability and ENSO, AMO and IPO [Camargo and Sobel, 2005; Nogueira and Kim, 2007; Mahala et al., 2015; Zhao et al., 2018]. In this study we analyzed all off-season TCs and our results differ from those that have found no trend in overall TC frequency, since we found decadal increasing trends in the NA and EP basin in both the pre and post-satellite eras. The results presented here suggest that climate change trends like higher SSTs and more favorable moisture environments (CC) between the months of Dec to May in the NA and EP basins seem to be the major factors behind decadal increasing trends in off-season TCs."**

2) Trends in climate indices: It is good that you added the table, but you now need to discuss these additional results in the text. Just stating that things are similar is not enough. What can we learn from the difference? I do not understand your explanation of the negative ENSO index in Figs. 3b and 5b and wonder whether there may be a mistake.

Thank you for the suggestion. We have now discussed similarities and differences between the original and detrended MLR models in the text and we have also added this sentence in the new version of the manuscript:

**When the MLR results of the original and detrended series are compared (Table 3), we find that the models with the detrended series exhibit lower R squares than the MLR models with the original series, yet those models were still found to be statistically significant which suggests that the correlation between off-season TCs and climate change factors is strong even after decadal trends are removed.**

Sorry for not elaborating a little bit more on why the ENSO seems to be mostly negative. Let me show you how the decadal average was calculated and that will help. We calculated the ENSO 3.4 decadal average by first calculating an average Dec-May ENSO 3.4 for each year and then we calculated the decadal average for each decade from that Dec-May ENSO 3.4 average for each year. Below we see the average Dec-May ENSO 3.4 for each year and when we calculate the decadal average, we get -0.327. We repeat that process with the other decades, and we get the ENSO 3.4 decadal average for each of the decades.

| Year | ENSO 3.4 |
|------|----------|
| 1900 | 0.4639176 |
| 1901 | -0.4926524 |
| 1902 | 0.8448318 |

| 1903 | -0.5903594 |
|------|------------|
| 1904 | -0.6698176 |
| 1905 | 0.8433211 |
| 1906 | -0.5974004 |
| 1907 | -0.6810894 |
| 1908 | -0.967404 |
| 1909 | -1.426191 |

3) Pre-satellite era: Again, it is good that you included the values in the table but now you need to discuss these findings in the text. You state in your reply that the MK test does not work for 60 years only. If that is the case, you should not present the results like this. Either use a different test or break down your timeseries in pentads instead. How is it possible that WP has negative trends in pre and post but positive in off? Please carefully check all numbers!

Sorry for the mistake in the WP off-season, the value was supposed to be a negative and not a positive. I also wanted to clarify something, it is not that the MK test doesn't work for 60 years it's that in the case of this study those 60 years are grouped into 6 decades, so when we do the test for the post-satellite era we are actually doing the test for 6 decades and not 60 years. In the new manuscript I have added a couple of sentences discussing the MK results for the post satellite era.

4) Data homogenization: The text you added creates at least some awareness of the problem. I think that should suffice for the moment.

Thank you.

5) Tropical depressions: I understand your point, but you should include this justification in the paper! You could state explicitly that it would be desirable to compare results including and excluding tropical depressions. Given the many uncertainties you are dealing with I suggest that you add 1-2 sentences to the conclusions with caveats (worse data quality in pre-satellite era, question of applying data homogenization for North Atlantic to other basins, lack of information about intensity).

I apologize for not including this in the previous manuscript. In the new version of the manuscript we have added a paragraph on limitations in the conclusion, here is the paragraph:

*"One of the main limitations of this work was the inclusion of tropical depressions in the off-season TC analysis. If data on TC intensity were widely available for all off-season TCs, it would have been possible to exclude weaker tropical depressions from the analysis since the detection and classification of those storms was more difficult in the pre-satellite era. Other limitations of this study include the issues of worst data quality in the pre-satellite era, the problem of applying a universal missed TC adjustment method to all basins analyzed and the lack of information on TC intensity for many storms, especially in the pre-satellite era. "*

6) TC tracks and SST change: Here you did not answer my question. Were there any changes over time in track or genesis location? Please check and include a respective statement in the text.

Sorry for that, based on spatial analysis looking at the mean center of the track and genesis location no major changes were detected. Statements regarding this have been added to the latest manuscript.

7) Relationship to cloud cover: The sentence you added is good, I think. There is, however, a misspelling in the reference (Klotzbach) here and elsewhere. Maybe just use "better" instead of "more optimum"?

Thank you for letting us know, we have corrected those errors.

8) ICOADS: What I meant is that you should explain how the ICOADS cloud estimates are produced and then discuss limitations (changes in data availability and methods to correct for that). The quote from Freeman et al. (2016) can help you to do this.

Thank you for being more specific as to what is it that you want to see in this section. We have added these sentences that explain how cloud estimates from ICOADS are produced and the limitations associated with the estimates:

[revised manuscript text omitted]

Landsea, C.W.,  Harper, B.A.,  Hoarau, K., and  Knaff, J.A.:  Can we detect trends in extreme tropical cyclones? *Science 2006*, **313**:452–454. DOI: 10.1126/science.1128448, 2006.

Landsea, C.W.:  Counting Atlantic tropical cyclones back to 1900. Eos, Transactions American Geophysical Union, 88(18), 197-202. https://doi.org/10.1029/2007EO180001, 2007.

Mahala, B. K., Nayak, B. K., and Mohanty, P. K.:  Impacts of ENSO and IOD on tropical cyclone activity in the Bay of Bengal. *Natural Hazards*, *75*(2), 1105-1125. https://doi.org/10.1007/s11069-014-1360-8, 2015.

Mann, H.B.:  Nonparametric tests against trend, Econometrica, 13, 245-259, 1945.

Mann, M.E.,  Sabbatelli, T.A.,  Neu, U.:  Evidence for a modest undercount bias in early historical Atlantic tropical cyclone counts. Geophys. Res. Lett., 34: L22707; doi:10.1029/2007GL031781, 2007.

Mann, M.E.,  Woodruff, J.D.,  Donnelly J., and  Zhihua, Z.:  Atlantic hurricanes and climate over the past 1,500 years. Vol. 460 (13 August 2009); doi:10.1038/nature0821, 2009.

McLeod, A. I.:  Kendall rank correlation and Mann-Kendall trend test. R Package Kendall, 2005.

Moon, I. J., Kim, S. H., and Chan, J. C.:  Climate change and tropical cyclone trend. *Nature*, *570*(7759), E3-E5, 2019.

Nogueira, R. C., Keim, B. D., Brown, D. P., and Robbins, K. D.:  Variability of rainfall from tropical cyclones in the eastern USA and its association to the AMO and ENSO. *Theoretical and applied climatology*, *112*(1-2), 273-283. https://doi.org/10.1007/s00704-012-0722-y, 2013.

Saunders, M., Lea, A.: Large contribution of sea surface warming to recent increase in Atlantic hurricane activity. *Nature* 451, 557–560,  doi:10.1038/nature06422, 2008.

Trenberth, K.E. and  Shea, D.J.:  Atlantic hurricanes and natural variability in 2005. *Geophys. Res. Lett.,* 33: L12704, doi:10.1029/2006GL026894, 2006.

Trenberth, K. E.:  Warmer oceans, stronger hurricanes. Scientific American, 297(1), 44-51. https://www.jstor.org/stable/26069374, 2007.

Trenberth, K.E., Zhang, R., & National Center for Atmospheric Research Staff (Eds). Last modified 10 Jan 2019. "The Climate Data Guide: Atlantic Multi-decadal Oscillation (AMO)." Retrieved from https://climatedataguide.ucar.edu/climate-data/atlantic-multi-decadal-oscillation-amo, 2019.

Venables, W. N. and Ripley, B. D.  Modern Applied Statistics with S. Fourth edition. Springer, 2002.

Walsh, K. J., Camargo, S. J., Knutson, T. R., Kossin, J., Lee, T. C., Murakami, H., Particular, C.: Tropical cyclones and climate change. Tropical Cyclone Research and Review, 8(4), 240-250, https://www.sciencedirect.com/science/article/pii/S2225603220300047, 2019.

Wang, C. and Lee, S. K.:  Global warming and United States landfalling hurricanes. Geophysical Research Letters, 35(2). https://doi.org/10.1029/2007GL032396, 2008.

Woodruff, S. D., Worley, S. J., Lubber, S. J., Ji, Z., Eric Freeman, J., Berry, D. I., ... & Wilkinson, C.: ICOADS Release 2.5: extensions and enhancements to the surface marine meteorological archive. *International journal of climatology*, *31*(7), 951-967, 2011.

Zhao, J., Zhan, R., Wang, Y., and & Xu, H.: (2018). Contribution of the interdecadal Pacific
oscillation to the recent abrupt decrease in tropical cyclone genesis frequency over the western
North Pacific since 1998. Journal of Climate, 31(20), 8211-8224. https://doi.org/10.1175/JCLI-
D-18-0202.1, 2018.

---

## Author Response (AR5)

Comments to the Author:

Thanks for addressing this last round of comments. I have only a few more minor points and then we can finally publish this:

L192: this technique

Fixed

L288: Klotzbach

Fixed

L342: correlation

Fixed

L391: a non-statistically significant

Fixed

L483: worse

Fixed

L534: Landseer should be Landsea

Fixed

Comments to the Co-Editor:

The authors of this manuscript appreciate all of the time that the Co-Editor and Referees have dedicated to reviewing this manuscript. Your suggestions and comments made this manuscript a much better paper.

[revised manuscript text omitted]